# Rift Valley fever outbreak among animal handlers in Kinyogoga Village, Nakaseke District, Central Uganda: A case study

Mariam Komugisha[1]*, Brian Kibwika[1], Benon Kwesiga[1], Richard Migisha[1], David Muwanguzi[2], Stella Lunkuse[2], Joshua Kayiwa[2], Hildah Tendo Nansikombi[1], Lilian Bulage[1], Dickson Stuart Tayebwa[3], Luke Nyakarahuka[4,5], Alex Riolexus Ario[1]

**1** Uganda Public Health Fellowship Program-Uganda National Institute of Public Health, Kampala, Uganda, **2** Ministry of Health, Kampala, Uganda, **3** Department of Veterinary Pharmacy Clinical and Comparative Medicine, Makerere University Kampala, Kampala, Uganda, **4** Uganda Virus Research Institute, Entebbe, Uganda, **5** Department of Biosecurity, Ecosystems and Veterinary Public Health, Makerere University, Kampala, Uganda

* mkomugisha@uniph.go.ug

## Abstract

Rift Valley fever (RVF) is a viral zoonosis which occurs sporadically in Uganda. On July 24, 2023, a 24-year-old male animal husbandry officer from Nakaseke District presented to a hospital in Kampala District with history of intermittent nosebleeds. He tested positive for Rift Valley fever virus (RVFV) by reverse transcriptase polymerase chain reaction (RT-PCR). We investigated to identify the source, estimate the magnitude, and determine the drivers of the outbreak, in order to inform control measures. We defined a suspected case as a sudden onset of fever (≥38°C) and ≥2 of: headache, skin rash, muscle/joint pain, intense fatigue, dizziness, coughing, abdominal pain, or unexplained bleeding from any site in a resident of Kinyogoga Village, Nakaseke District from June 1–July 30, 2023. A confirmed case was a suspected case with positive RT-PCR for RVFV. We conducted active case finding, environmental assessments, interviewed secondary cases and the family members of the deceased index case, and collected blood samples for testing. We identified eight case-patients, all males and one died. Median age for both the suspected and confirmed case-patients was 25 years (range: 19–28). Symptoms included: high fever (100%), headache (100%), vomiting blood and nose bleeding (25%). The case-patients identified (one animal husbandry officer and seven herders) worked on three affected farms that had recently recorded high rates of abortion and mortality in cattle and shoats. The index case started vomiting blood on July 19, 2023; RVF was confirmed five days later. Prior to that, he had visited three health facilities within the outbreak area that clinically diagnosed brucellosis, typhoid and gastritis. This RVF outbreak likely resulted from contact with infected livestock and their products. We recommend training healthcare workers, animal health workers and animal handlers

**Data availability statement:** All relevant data are within the paper and its Supporting Information files.

**Funding:** The project was supported by the President's Emergency Plan for AIDS (PEPFAR) through the United States Centers for Disease Control and Prevention Cooperative Agreement number GH001353-01 through Makerere University School of Public Health to the Uganda Public Health Fellowship Program, Ministry of Health. The findings and conclusions in this report are those of the author(s) and do not necessarily represent the official views of the US Centers for Diseases Control and Prevention, the Department of Health and Human Services, the Agency for Toxic Substances and Disease Registry, Makerere University School of Public Health, or the Ministry of Health. The funders had no role in study design, data collection and analysis, decision to publish, or preparation of the manuscript.

**Competing interests:** The authors have declared that no competing interests exist.

on RVF, strengthening surveillance, vaccinating livestock, implementing mosquito control measures and conducting community education.

## Introduction

Rift Valley fever (RVF) is an acute viral hemorrhagic fever caused by Rift Valley fever virus (RVFV), a member of the genus *Phlebovirus* belonging to the family *Bunyaviridae* [1–3]. The virus can be transmitted from infected livestock (cattle, sheep, goats and camels) to humans through contact with blood, body fluids, or animal tissues during slaughtering or butchering, assisting with birth in livestock, conducting veterinary procedures, or from the disposal of carcasses or fetuses [4]. Transmission can also occur through bites from infected mosquitoes, most frequently the *Aedes* and *Culex* spp. mosquitoes, but person-to-person transmission has not been documented [5]. Persons in close contact with the livestock and their products such as herders, farmers, and animal health practitioners are at higher risk of infection [4]. In animals, RVFV is mainly transmitted by mosquitoes [4,6,7].

The infection has an incubation period of 2–6 days in humans, and those infected may remain asymptomatic, while others might develop mild or severe symptoms [6,8]. The clinical signs and symptoms are non-specific in humans. Those with mild form of RVF may experience signs and symptoms such as fever, sudden onset of flu-like signs, body weakness, joint pain, muscle pain, headache, loss of appetite, vomiting, confusion, neck stiffness, sensitivity to light and dizziness [9–11]. Clinical diagnosis can be difficult because RVF presents with symptoms that are similar to those of other endemic illnesses, such as malaria and brucellosis [11,12].

Symptoms of RVF usually last 4–7 days, after which Immunoglobulin M (IgM) antibodies can be detected, and the virus disappears from the blood. Immunoglobulin G (IgG) antibodies can also be detected afterwards and persist for several years [7,13]. People with severe form of RVF may experience hemorrhagic symptoms such as vomiting blood, passing blood in the faeces, a purpuric rash, bleeding from the nose or gums and bleeding from venipuncture sites [5,10]. The overall case fatality rate of RVF is < 1%. However, in patients with hemorrhagic form, the case-fatality rate is up to 50% [4,14].

RVFV infects domestic ruminants such as cattle, sheep, and goats in an age-dependent manner, where young animals are more susceptible than adults [15]. In livestock, the disease can cause increased abortions and stillbirths, and high mortality in neonates and weaners leading to significant economic losses [6]. There is no specific treatment for RVF in humans and livestock, but supportive care may prevent complications and decrease mortality [7,16]. Vaccination of livestock against RVF is the primary method for preventing RVF infection in endemic areas [17,18]. Both live and inactivated vaccines have been developed for use in livestock. However, vaccination against RVF in animals is not widely practiced in most countries [19]. In Uganda, two live attenuated RVF vaccines are commercially available. However,

RVF mass vaccination in livestock has not yet been implemented partly because RVF remains a neglected zoonotic disease [20].

RVFV was first isolated during an epidemic among sheep in the Rift Valley in Kenya in 1931 [21]. Between 2000–2016, widespread outbreaks have occurred in various countries such as Kenya, Tanzania, Somalia, Sudan, Niger, Madagascar, South Africa, Saudi Arabia and Yemen [22–29]. In Uganda, the first RVF human case was identified in 1968 followed by an outbreak that occurred in Kabale District in 2016 [30]. Uganda has continued to experience an increase in sporadic outbreaks of RVF. From 2016 to 2018, ten independent outbreaks occurred in ten districts [31]. The disease is endemic in several regions of the country, a recent study reported a seroprevalence of 11% in cattle, sheep and goats, country-wide [32]. According to the National Public Health Emergency Management database at the Ministry of Health, RVF was confirmed in 21 districts country-wide from 2017–2023.

On July 24, 2023, the Uganda National Public Health Emergency Operations Center (PHEOC) was notified of a suspected viral hemorrhagic fever (VHF) case-patient by a hospital in Kampala District through the Event-Based Surveillance (EBS) unit. On the same day, the case-patient tested positive for RVF using reverse transcriptase polymerase chain reaction (RT-PCR) at the Central Public Health Laboratory (CPHL). We investigated to identify the source, estimate the magnitude, and determine the drivers of the outbreak to inform control and prevention measures for future outbreaks.

## Materials and methods

### Ethics statement

This study was conducted as a response to public health emergency by the National Rapid Response Team. The Ministry of Health Uganda provided administrative clearance to conduct this investigation. In addition, we received a non-research determination clearance from the US Centers for Disease Prevention and Control (US CDC). This activity was reviewed by CDC and was conducted consistent with applicable federal law and CDC policy. § §See, e.g., 45 C.F.R. part 46, 21 C.F.R. part 56; 42 U.S.C. §241(d); 5 U.S.C. §552a; 44 U.S.C. §3501 et seq. We obtained verbal informed consent from secondary case-patients, and family members of the deceased index case. We obtained verbal informed consent in the local language (Luganda). Written consent could not be obtained because it was impractical as most of the participants were illiterate. However, we explained to all participants the purpose of the investigation. Participants were also informed that their participation was voluntary and their refusal to answer any or all of the questions would not result in any consequences. We conducted interviews in privacy to ensure confidentiality, and the data was kept under password protection by the study team.

### Outbreak area

This outbreak occurred in Kinyogoga Village located in Kinyogoga Subcounty, Nakaseke District. The district is located in Central Uganda (Fig 1), and has an estimated population of 254,900 living in 8 sub-counties [33]. Nakaseke District is located within Uganda's cattle corridor—a semi-arid rangeland belt that stretches from the southwest to the northeast of the country, encompassing over 30 districts. This corridor is the backbone of Uganda's livestock industry, supporting the highest concentrations of cattle, goats, and sheep in the country. Nakaseke District alone hosts an estimated 250,000 cattle, alongside substantial populations of goats and sheep. Livestock rearing is the main economic activity in Nakaseke District, contributing to Uganda's national output of milk, meat, and hides, and making it central to both food security and rural livelihoods.

Fig 1 was drawn using qgis version 3.22 and the district, subcounty and village shape files from Uganda Bureau of Statistics (UBOS) (Shape file source: https://ubosgis.ubos.org/portal/home/item.html?id=36a67058df7940e4a1ec284a-daebd16d). The polygon data for the figure can be used without restrictions (https://ubosgis.ubos.org/portal/home/item.html?id=36a67058df7940e4a1ec284adaebd16d). The base layer for the map was obtained from within qgis version 3.22. qgis is an open-source software or open license. We linked subcounty shape files from UBOS to the base layer within qgis; How to link shape files using vector files in qgis: https://docs.qgis.org/3.22/pdf/en/QGIS-3.22-DesktopUserGuide-en.pdf

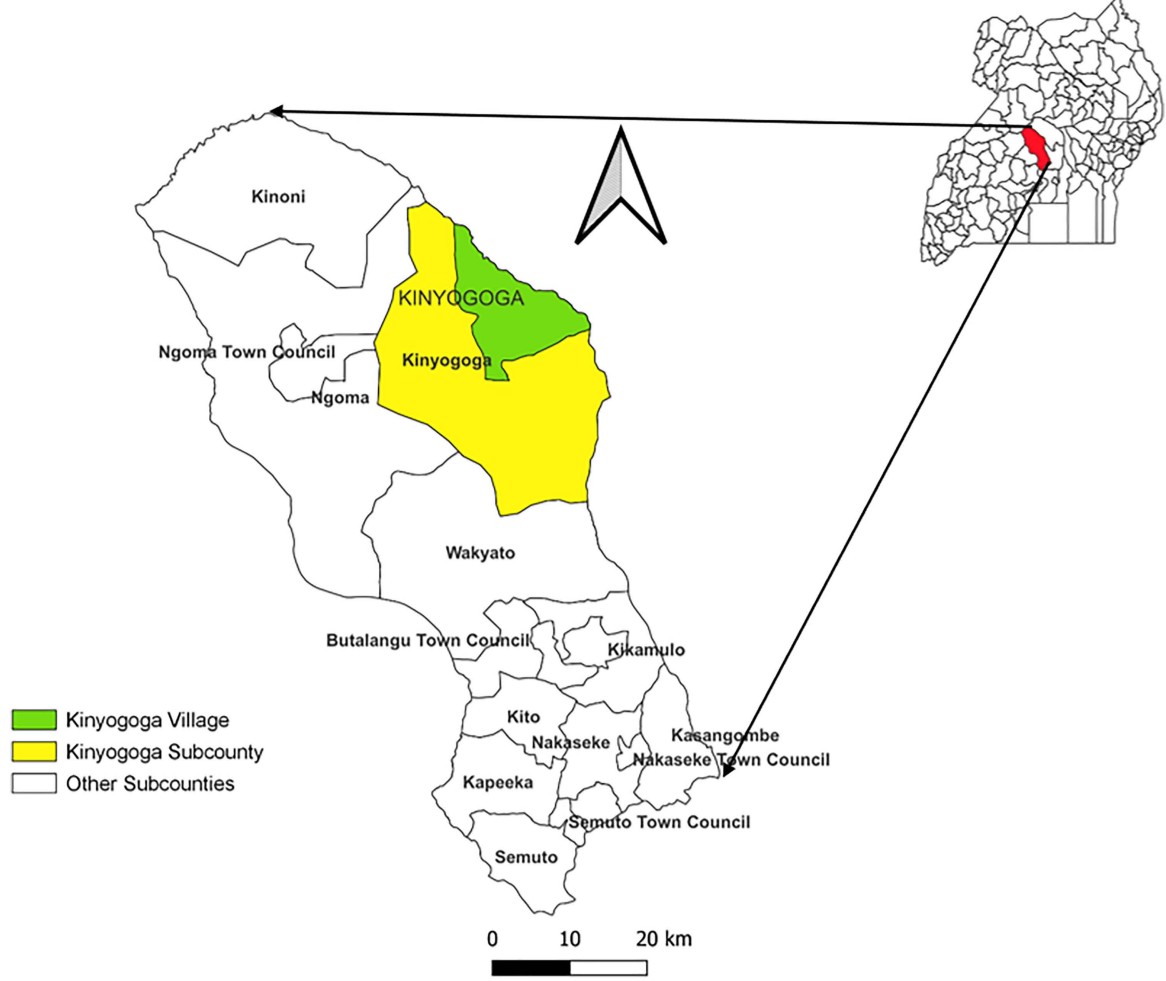

**Fig 1. Location of Kinyogoga Village in Kinyogoga Subcounty, Nakaseke District, Uganda.**

## Case definition and finding

We defined a suspected case as an individual with the sudden onset of fever (≥38°C), with no clear alternative diagnosis, and at least two of the following signs and symptoms: headache, skin rash, muscle/joint pain, intense fatigue, dizziness, coughing, abdominal pain, or unexplained, bleeding from any site from June 1–July 30, 2023, in a resident of Kinyogoga Village, Nakaseke District. A confirmed case was defined as a suspected case with positive RT-PCR for RVFV from June 1-July 30, 2023.

To identify additional cases, the team visited several health facilities where the index case sought care; we reviewed records and interviewed health workers. We interviewed index case family members and herders residing or working on farms which had reported high rates of abortions and mortality of cattle and shoats within the past one month. We used snowballing to identify additional cases who presented with similar signs and symptoms in the community. We interviewed case-patients to determine the date of RVF onset, signs and symptoms, demographics, and interactions with livestock and their products.

## Descriptive epidemiology

We interviewed secondary cases and the family members of the index case about their demographics, animal-related activities, number of animal abortions and deaths, the vaccination status of the animals, and practices that might have

exposed to RVFV. The proportion of case-patients were calculated for sex, occupation, and symptoms. We summarized age of the case-patients using range and median. We constructed an epidemic curve to describe the distribution of RVF cases by time of symptom onset. We used Quantum Geographic Information System (QGIS, version 3.22) to draw maps showing the outbreak area and the distribution of the affected farms in Kinyogoga Village, Nakaseke District.

### Environmental assessment

During the case-finding activities, we conducted onsite inspections of the three farms in Kinyogoga Village to observe breeding sites for mosquitoes and exposure risk behaviors among persons.

### Laboratory investigations

We collected blood samples from seven secondary cases. Samples were transported to the Uganda Virus Research Institute (UVRI) laboratory in Entebbe, Uganda. Reverse transcriptase polymerase chain reaction was performed to confirm RVF as previously described [34]. Briefly, we extracted RNA from human samples using MagMAX magnetic bead system (Life Technologies, Carlsbad, CA) following the manufacturer's instructions. We mixed 100 µL of each specimen with 400µL of lysis buffer supplemented with 2µL of carrier RNA solution. Extraction was performed using the BeadRetriever automated magnetic bead separation system, and RNA was eluted in 90µL of elution buffer (Applied Biosystems, Inc., Waltham, MA). We used the following primer and probe set: 5' -TGAAAATTCCTGAGACACATGG-3'(RVFL-2912fwdGG), 5'-ACTTCCTTGCATCATCTGATG-3'(RVFL-2981revAC), and FAM-5'-CAATGTAAGGGGCCTGTGTGGACTTGTG-3'-BHQ (RVFL-probe-2950). We cycled the samples on an ABI Quant Studio (ABI) under the following conditions: one cycle of 51°C for 30 minutes and 94°C for 2 minutes, followed by 40 cycles of 94°C for 15 seconds, 56°C for 30 seconds, and 68°C for 2 minutes, with a final extension at 68°C for 5 minutes.

## Results

### Descriptive epidemiology

We identified one index case and seven secondary RVF cases from Kinyogoga Village; the index case died. Two of the case-patients including the index case, were RT-PCR-positive. All case-patients were male with a median age of 25 years (range:19–28 years). Seven (88%) case-patients were herders and one (12%) was an animal husbandry officer (S1 Data). Symptoms included: high fever (100%), headache (100%), abdominal pain (63%); 25% experienced vomiting blood and nose bleeding (Fig 2). Onset of symptoms among case-patients started from June 17, 2023, to July 26, 2023. The index case died 16 days after the first symptom onset. Concurrent with the human cases, there were increasing reports of abortions and deaths in livestock which started in early June 2023.

The index case was a 25-year-old male animal husbandry officer. He developed initial symptoms including fever, headache and abdominal pain on July 12, 2023. He sought care at a Health Centre 3 (mid-level public health facility) on July 12, 2023, and was treated for gastritis. He further sought care at two other health facilities during July 16–19, 2023, where he was managed for typhoid and brucellosis. On July 19, a week since symptom onset, he started experiencing nose bleeding and vomiting of blood. He self-referred to a hospital in Kampala District where a viral hemorrhagic fever (VHF) was suspected; he tested positive for RVFV on July 23, 2023. He died on July 28, 2023, five days later.

All case-patients resided or worked on three farms that had experienced abortion storms and high mortality of livestock, particularly among calves, lambs and kids in Kinyogoga Village. The affected farms were neighboring each other in the same village (Fig 3).

Fig 3 was drawn using qgis version 3.22. The village shape files for the figure has been sourced from the humanitarian data exchange dataset (https://data.humdata.org/dataset/uganda-villages) and can be used under an open license (https://data.humdata.org/dataset/uganda-villages). The base layer for the map was obtained from within qgis version 3.22.

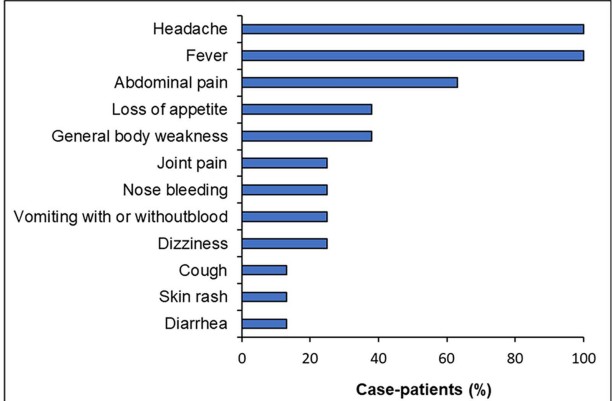

**Fig 2. Symptoms presentation of case-patients during Rift Valley fever outbreak in Nakaseke District, Uganda, June–July 2023.**

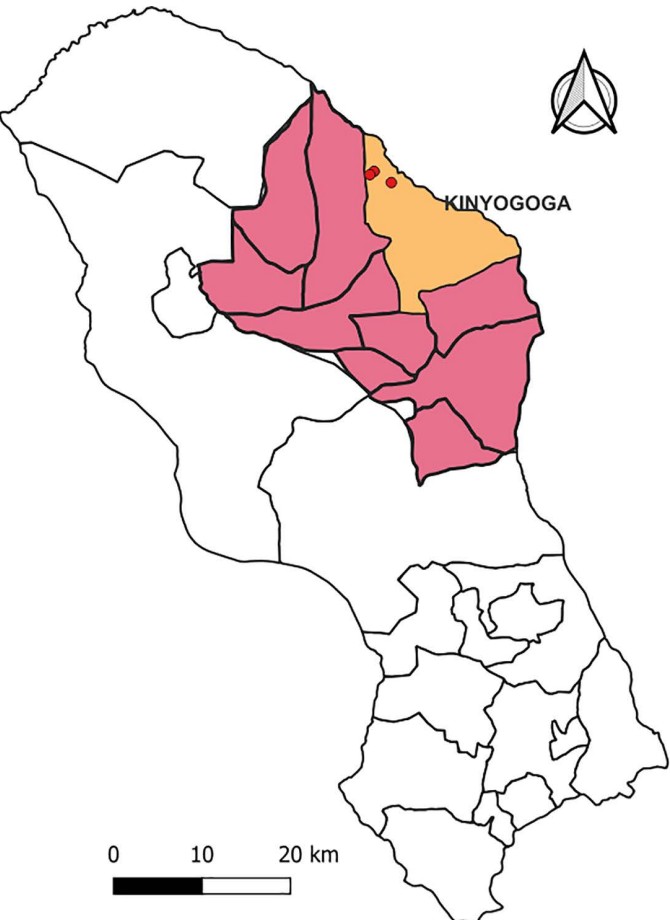

**Fig 3. Map of Nakaseke showing location of farms affected by Rift Valley fever (red dots) in Kinyogoga Village, Kinyogoga Subcounty, June–July 2023.**

qgis is an open-source software or open license. We linked village shape files to the base layer within qgis; How to link shape files using vector files in qgis: https://docs.qgis.org/3.22/pdf/en/QGIS-3.22-DesktopUserGuide-en.pdf

From the information gathered, the abortion storms and livestock deaths began early June and continued throughout the month of July 2023; human cases started to appear on June 17, 2023. The case that developed onset of symptoms in June had mild symptoms, and he was detected during the outbreak investigation in the month of July. On July 12, the index case started experiencing RVF symptoms. The index case (animal husbandry officer) had conducted a postmortem on an animal that died from an unknown disease on July 5, 2023; he developed symptoms after 7 days. Secondary cases rapidly increased and peaked on July 15, 2023 (Fig 4). The other case patient who tested positive for RVFV was a 28-year-old herdsman who developed RVF related symptoms such as fever, headache, vomiting, joint pain, and skin rash on July 21, 2023. He sought care from a clinic in Nakaseke District where he was diagnosed with typhoid and brucellosis.

## Environmental assessment

The case-patients worked on the three affected farms where cattle, sheep and goat herds shared grazing areas and watering points. The affected farms had water dams which acted as breeding sites for the mosquitoes. All case-patients participated in the slaughter and butchering of sick animals, assisting animal birth, handling aborted animal fetuses and retained placenta without wearing personal protective equipment (PPE). In addition, they consumed meat from livestock found dead and regularly milked animals and drank raw milk. The index case (animal husbandry officer) collected blood samples from sick animals and conducted a post-mortem on a sick animal prior to the onset of his illness. The investigation team educated the secondary cases and the family of the deceased index case about RVF, covering its cause, transmission, risk factors as well as control and prevention measures.

## Discussion

An outbreak of Rift Valley Fever (RVF) occurred in Kinyogoga Village, Nakaseke District, between June and July 2023. The outbreak involved two confirmed cases, both male: one was an animal husbandry officer, and the other was a livestock herder.

One of the confirmed case-patients developed mild symptoms, while the other developed severe RVF infection, had a delayed diagnosis and died.

Overall, there was a delay in diagnosing RVF. The index case who died initially presented with nonspecific symptoms such as high fever, headache and abdominal pain. He sought care at four health facilities before a VHF was suspected by the healthcare workers after 12 days of seeking health care. The two confirmed cases were initially misdiagnosed with brucellosis and typhoid. The early-disease symptoms are non-specific, making it difficult to differentiate RVF from other

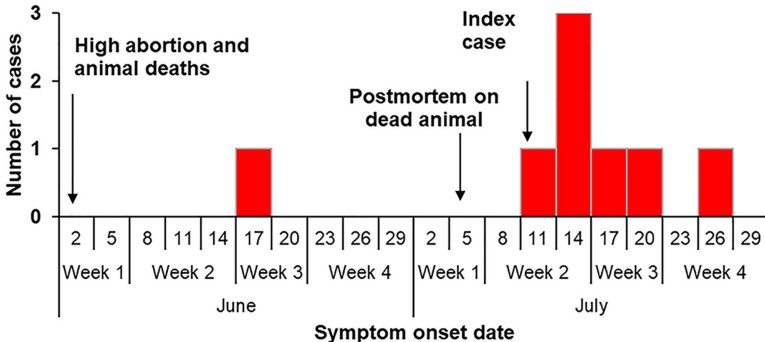

**Fig 4. Distribution of symptom onset dates of Rift Valley fever case-patients: Nakaseke District, Uganda, June-July 2023.**

endemic diseases such as typhoid and brucellosis [11]. For diseases with non-specific symptoms, the availability of diagnostic facilities is essential to enable timely detection of cases of an emergent epidemic threat [35]. Unfortunately, health facilities in remote areas like Nakaseke District have limited diagnostic capacity, increasing the likelihood of misdiagnosis and delay in outbreak detection [36]. In fact, RVFV diagnosis in Uganda is currently restricted to the National reference laboratories located in Kampala, the capital. Alternatively, the misdiagnosis could have resulted from limited knowledge and low suspicion index among health workers. Higher levels of suspicion and laboratory diagnosis can help in the early detection of the disease, ensure timely treatment, and recovery of the infected persons. Our findings highlight the need to improve access to diagnostic services and educating healthcare workers about VHFs in particular RVF may facilitate timely detection and appropriate management of persons with conditions such as RVF.

The case-patients were animal handlers who engaged in slaughtering and butchering sick animals, assisting animal birth, handling aborted animal fetuses and retained placenta in ways that allow the RVFV transmission. Studies have shown that contact with aborted animal tissue [37,38], and slaughtering an animal [39–41] and birthing an animal [37] increased the risk of RVFV exposure. The case-patients performed multiple risk activities without using PPEs, this likely increased the risk of RVFV exposure. Studies have shown that the use of PPE and disinfectants may help to reduce the risk of transmission [37].

The affected farms had a watering point which acted as a breeding site for the mosquitoes. There is also a possibility that case-patients particularly herders got exposed to RVFV through bites from infected *Aedes* and *Culex* spp. mosquitoes [20]. The affected farms where the case-patients worked and resided had water ponds for water catchment, which can serve as mosquito breeding sites. Stagnant water has shown previously to facilitate malaria outbreaks through increase in mosquito breeding sites [42], and may similarly have provided breeding sites for RVF-carrying vectors in this area. Control of mosquito breeding sites and protection against mosquito bites may help prevent the spread of RVF.

The outbreak affected only males and young adults aged 19–28 years. In livestock rearing communities, males play gender-based roles such as grazing and slaughtering of animals which increases the risk of zoonotic disease transmission from animals to males compared to females [43]. In addition, compared to children and older persons, young adults have frequent participation in livestock-related activities such as grazing, slaughtering, and butchering of animals [11]. Sensitization programs to enhance exposure prevention strategies for RVF should target males and young adults.

The animals on the affected farms were not vaccinated against RVF. Previous studies have suggested that livestock vaccination against RVF during an outbreak could be the most effective control measure for RVF and eliminate one of the main sources of human infection [16,44]. Despite the reports of the availability of the live attenuated RVF vaccines in the Ugandan market, the vaccination of animals against RVF is limited [20]. The lack of animal vaccination is likely due to RVF being a neglected disease and not considered a national priority. Additionally, because the disease occurs sporadically, farmers may have paid little attention to it despite its zoonotic potential. The Ministry of Agriculture and the Ministry of Health, working collaboratively under the One Health approach, need to strengthen awareness programs for RVF and other related VHFs. The outbreak was initially detected in humans, though reports suggest it originated in animals before spreading to humans. This study underscores the need for enhanced surveillance of RVF in animals to detect outbreaks before they spill over into human populations. Additionally, once diseases with a high zoonotic risk are identified, a One Health approach should be implemented to ensure early detection and a coordinated, effective response to RVF.

## Study limitations

There is a likelihood of underestimation of the true magnitude of the outbreak during the investigation based on the possibility of having missed cases who exhibited mild or no symptoms. The authors identified the exposure factors for RVF infection, however, due to the limited number of cases, the authors could not conduct statistical analysis to ascertain risk factors for transmission. In this study, we reported the one death that occurred during the outbreak, the authors could not determine the Case Fatality Rate due to the fewer number of case-patients identified. Animal samples were not collected during the outbreak due to logistical concerns.

## Conclusion

This 2023 RVF outbreak in Kinyogoga, Nakaseke District, Central Uganda affected livestock herders and an animal husbandry officer. Compelling evidence suggests the disease originated from contact with infected livestock and was likely caused by occupational exposure such as herding and clinical handling of animals, or through contact with and consumption of infected livestock products. The suspicion index for VHFs is low among healthcare workers, and the animal disease surveillance system is inadequate, impending timely detection of RVF outbreak. We recommend training of healthcare workers, animal health workers and animal handlers on RVF, strengthening surveillance in both animals and humans, vaccinating animals against RVF, implementing vector control measures and conducting community education on risk factors and prevention measures against RVF.

## Supporting information

**S1 Data. Rift Valley fever outbreak among animal handlers in Kinyogoga Village, Nakaseke District, Central Uganda.**
(XLSX)

## Acknowledgments

We extend our appreciation to Nakaseke District team for the technical and administrative support they offered during the investigation. Our appreciation goes to the participants for providing all the necessary information during the investigation. We appreciate the Uganda Virus Research Institute for testing all samples and the timely release of laboratory results.

## Author contributions

**Conceptualization:** Mariam Komugisha, Alex Riolexus Ario.

**Data curation:** Mariam Komugisha.

**Formal analysis:** Mariam Komugisha.

**Funding acquisition:** Alex Riolexus Ario.

**Investigation:** Mariam Komugisha, Benon Kwesiga, Joshua Kayiwa.

**Methodology:** Mariam Komugisha, Brian Kibwika, Benon Kwesiga, Joshua Kayiwa.

**Supervision:** Benon Kwesiga, Richard Migisha, Hildah Tendo Nansikombi, Lilian Bulage, Alex Riolexus Ario.

**Writing – original draft:** Mariam Komugisha.

**Writing – review & editing:** Brian Kibwika, Benon Kwesiga, Richard Migisha, David Muwanguzi, Stella Lunkuse, Hildah Tendo Nansikombi, Lilian Bulage, Dickson Stuart Tayebwa, Luke Nyakarahuka, Alex Riolexus Ario.

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
