## [Decision Letter · Decision Letter 0]

10 Sep 2024

PGPH-D-24-01296

Rift Valley fever outbreak among animal handlers linked to contact with infected animal products in Kinyogoga Village, Nakaseke District, Central Uganda, June–July 2023

Dear Dr. Mariam Komugisha,

Thank you for submitting your manuscript to PLOS Global Public Health. After careful consideration, we feel that it has merit but does not fully meet PLOS Global Public Health’s publication criteria as it currently stands. Therefore, we invite you to submit a revised version of the manuscript that addresses the points raised during the review process.

We look forward to receiving your revised manuscript.

Kind regards,

Sukanta Chowdhury, Ph.D

Academic Editor

Journal Requirements:

1. Your funding information on the submission form indicates, Grant Recipient Dr Alex Riolexus Ario. Please indicate by return email the full and correct funding information for your study and confirm the order in which funding contributions should appear. Please be sure to indicate whether the funders played any role in the study design, data collection and analysis, decision to publish, or preparation of the manuscript.

2. Please provide separate figure files in .tif or .eps format.

3. In the online submission form, you indicated that "the data sets can be made available upon reasonable request from the corresponding author". 

3. Uploaded as supplementary information.

4. Figure 1 and 2: please (a) provide a direct link to the base layer of the map (i.e., the country or region border shape) and ensure this is also included in the figure legend; and (b) provide a link to the terms of use / license information for the base layer image or shapefile. We cannot publish proprietary or copyrighted maps (e.g. Google Maps, Mapquest) and the terms of use for your map base layer must be compatible with our CC-BY 4.0 license. 

Reviewers' comments:

Reviewer-1

Authors Komugisha et al. describe a very specific RVFV outbreak in Uganda with human cases. Description of the clinical impacts of this outbreak focus on only two of the few cases (N = 8), as these were the ones they could confirm via RT-PCR. No other diagnostic testing to confirm exposure (such as serological testing) was utilized, therefore the true scope of this outbreak is not described by this paper. Similarly, animal testing is not described. While this paper is well written, it suffers from redundancy and does not investigate beyond the surface. I would not describe this as a research article, and may be better presented as a case study.

Specific comments and edits:

1. This paper is light on references, especially in the introduction.

2. Page 3, line 62: should read "the infection" rather than "the disease has an incubation period". The disease is symptomatic presentation resulting from infection. Rather, the virus has an incubation period before the symptoms commence.

3. Page 3, lines 67 and 68: IgM is not defined, but Immunoglobulin G is fully spelled out without abbreviation. Review for consistency.

4. Page 4, lines 74 and 80: Rift Valley Fever Virus has already been defined previously, and should therefore only be referred to as RVFV following initial mention.

5. Page 4, paragraph starting on line 74: No mention of vaccines for livestocks is included, and this feels like a major oversight. If the vaccines are not used or not available to farms and herds in this region, this should be discussed. In fact, there is no mention of animal vaccination at all in the entire paper!

6. Figure 1: check map for text clarity, it prints pixelated for me in it's submitted size/resolution.

7. Page 6, section on descriptive epidemiology is lacking detail, so it is impossible to assess the actual work done beyond basic arithmetic to calculate percentages. No statistics.

8. Pages 6 and 7, ethical considerations section causes concern. No published consent forms or questions, no discussion of avoiding coercion while obtaining strictly verbal consent. What language was used? No discussion of who was not included, or if any individuals were approached and declined. Just says "all respondents were above the age of 18 and gave verbal consent for interviews." Using COVID as a reason for only obtaining verbal consent in 2023 is not convincing enough. Consenting practices are important.

9. Page 7, line 157: were participants asked about a single instance of fever or recurrent fevers? What about temperature of fevers? RVF typically is associated with high fever.

10. Page 7, line 160-161: "The time to death from symptom onset was 16 days." This section is confusing as currently written. Is this referring to the individual human that died, or the livestock mentioned in the previous sentence? Revise this section for clarity.

11. Figure 3: what are the green panels in this figure? Not described anywhere.

12. Page 9, lines 208-212: How many individuals had a differential diagnosis for brucellosis and typhoid? Both confirmed cases were initially misdiagnosed for these two infections, plus one mention of gastritis or gastric ulcer. Is this an issue with the clinical teams not being knowledgeable about RVFV, or not performing diagnostic testing? Further description is warranted in the discussion, because out of the two confirmed cases of this outbreak study, one person died, and the way it was described, it seems it was from medical negligence.

13. Page 10, line 216: I'm not sure it is valuable to compare basic percentages with such a small number of surveyed individuals and no further testing performed. The extremely small N is likely the contributing factor to the higher fatality rate compared to Sudan and all other outbreaks that weren't mentioned.

14. Page 10, line 219: mention of CCHF, was there any investigation to whether co-infection was possible here? Any reported co-infections that suggest this is a possibility? Concerns about serological diagnoses? This sentence seems like an afterthought.

15. Page 10, lines 221-225: again, no mention of vaccines for livestock, also failed to mention mosquito abatement.

16. Page 10, lines 238 - 240: mention of herdsmen having increased exposure is complicated because there are much more thorough longitudinal studies that show distinct roles in Kenya (Cook et al.), and butchery was the most risky, well above herding. This study fails to discuss the challenges distinguishing these roles in their specific population due to overlapping roles/behaviors. Would have liked to see a more thorough discussion on this and how performing multiple duties can increase risk of RVFV exposure.

17. Page 11, line 253: study limitations are an afterthought as well. This study has so many limitations that are not described. Why were individuals with no or mild symptoms excluded, when RVFV infection has a high rate of asymptomatic cases? How can the epidemiology of this outbreak be investigated if it is based on only two confirmed cases? Much of these findings seem like general assumptions.

18. Page 11, line 261 and beyond: why is the education factor only now being mentioned in this paper? No documentation or descriptions of methods used for educating are included. Do the authors plan to follow up at a later date to track efficacy of their educational efforts?

Reviewer-2:

General comments

These are valuable findings and raise awareness of the importance of having the human and animal health sectors work together to tackle zoonotic disease. In addition, of increasing awareness among clinical and veterinary personal of potential circulating pathogens, having robust surveillance systems, and having appropriate diagnostics in place for the timely and accurate detection of disease to reduce/prevent morbidity and mortality in both humans and animals.

I would recommend proof-reading the manuscript again for English grammar.

Abstract

Line 25: RVF virus

Line 26: lower case for “reverse transcriptase”

Line 27: “identify the source, estimate the magnitude, and …”

Line 32: RVF virus

Introduction

Line 67-68: Immunoglobulin (Ig) M…..IgG

Line 74: Rift Valley fever virus = RVFV, check throughout manuscript

Line 88: Which animals? Countrywide or a specific region?

Methods

Line 117: rates

Line 123: “The proportion of case-patients were calculated for sex, occupation and symptoms”

Line 133: herds men or herdsmen? – check throughout manuscript

Line 147: Is verbal consent sufficient in Uganda?

Results

Line 153: Be careful about reporting CFR with a small sample size, only a single death and 8 cases. Add to limitations.

Line 162: Rift Valley fever – check correct use of capitals throughout manuscript

Line 180: “…year, which started in early June and continued throughout…”

Line 182: When/how was the case detected? Perhaps rephrase the sentence.

Line 202: no comma after fetuses (also plural)

Line 204: “…and regularly milked…”

Line 206: post-mortems

Discussion

Line 216: Be careful about reporting CFR with a small sample size. Perhaps better to just describe the single case, that the delay in correct diagnosis may have contributed to this patients death. Add to limitations that the small sample size limits statistical analysis to identify risk factors.

Line 240: Aedes spp and Culex spp. (scientific names in italics)

Line 260: “…weak, impeding timely detection, which…”

---

## [Decision Letter · Decision Letter 1]

26 Dec 2024

PGPH-D-24-01296R1

Full title: Rift Valley fever outbreak among animal handlers in Kinyogoga Village, Nakaseke District, Central Uganda: a case study

Dear Dr. Mariam Komugisha,

Thank you for submitting your manuscript to PLOS Global Public Health. After careful consideration, we feel that it has merit but does not fully meet PLOS Global Public Health’s publication criteria as it currently stands. Therefore, we invite you to submit a revised version of the manuscript that addresses the points raised during the review process.

We look forward to receiving your revised manuscript.

Kind regards,

Sukanta Chowdury, Ph.D

Academic Editor

Editor Comments:

• Abstract: line 28. The sentence is not correct grammatically (A suspected case was a sudden onset of fever…….). You consider to write this way (We defined a suspected case as an individual with the sudden onset of fever (≥38°C)………….)

• The case definition is less clear to me. Did each suspected patient show all clinical signs including sudden onset of fever (≥38°C) and ≥2 of headache, skin rash, muscle/joint pain, intense fatigue, dizziness, coughing, abdominal pain, or bleeding? Or showed few clinical signs? Clarify the case definition in abstract and method section. You mentioned “bleeding”. Please specify whether it was nose bleeding/mouth bleeding/others…

• Median age for whom? Suspected cases or confirmed cases? Clarify

• All case-patients mean 8 cases or 2 cases? Clarify this in the abstract.

• hemorrhagic symptoms: where was it shown? Skin or conjunctiva?

• Diagnosed brucellosis, typhoid and gastritis through clinically or laboratory tests?

• Specify “infected animals’ products”. Meat/milk? What types of animals (livestock/wild)?

• Descriptive epidemiology section (method): why you mentioned case patients and index case separately? Case patients included index case. You can mention index case and secondary cases

• Why you included animal sickness history under environmental section? I have not seen any environmental variables. Revise this both method and result section.

• Add detail information about RT-PCR TaqMan assay

• Result: The fatality was the index case-patient (animal husbandry officer) who developed….Revise the sentence.

• Limitation: According to your study result, the CFR was 50%. Clarify it.

• Conclusion: The suspicion index for VHFs is low among clinicians…..this is not clear to me. Who are clinicians? Medical doctor or vet. Doctor?

• Animal disease surveillance system is weak…This is vague statement. Is there ongoing surveillance? Or inadequate surveillance. Revise this section.

• Training for clinician… why clinician? How about veterinarian/animal handler/quack?

Reviewers' comments:

Reviewer #1: The authors have done a fantastic job addressing all reviewer comments. This manuscript is much more clear after the revision, and highlights the important work conducted by the team of authors.

One small edit: Page 6, Line 128: Add a period at the end of the final sentence in this paragraph.

Reviewer #2: The authors have addressed my comments and the readability of the manuscript has greatly improved. My remaining concern is regarding the verbal informed consent. Generally, a witnessed mark or thumb-print is standard practice for consenting illiterate participants. Was verbal informed consent approved by an ethics committee?

Additional minor comments:

Check consistency of using "RVF virus" vs "RVFV"

Line 196: should be lambs not lamb

---

## [Editor Report · Decision Letter 2]

7 Jul 2025

Full title: Rift Valley fever outbreak among animal handlers in Kinyogoga Village, Nakaseke District, Central Uganda: a case study

PGPH-D-24-01296R2

Dear Dr. Mariam Komugisha,

We are pleased to inform you that your manuscript 'Full title: Rift Valley fever outbreak among animal handlers in Kinyogoga Village, Nakaseke District, Central Uganda: a case study' has been provisionally accepted for publication in PLOS Global Public Health.

Best regards,

Sukanta Chowdury, Ph.D

Academic Editor
